# View-consistent Object Removal in Radiance Fields

## ABSTRACT

Radiance Fields (RFs) have emerged as a crucial technology for 3D scene representation, enabling the synthesis of novel views with remarkable realism. However, as RFs become more widely used, the need for effective editing techniques that maintain coherence across different perspectives becomes evident. Current methods primarily depend on per-frame 2D image inpainting, which often fails to maintain consistency across views, thus compromising the realism of edited RF scenes. In this work, we introduce a novel RF editing pipeline that significantly enhances consistency by requiring the inpainting of only a single reference image. This image is then projected across multiple views using a depth-based approach, effectively reducing the inconsistencies observed with per-frame inpainting. However, projections typically assume photometric consistency across views, which is often impractical in real-world settings. To accommodate realistic variations in lighting and viewpoint, our pipeline adjusts the appearance of the projected views by generating multiple directional variants of the inpainted image, thereby adapting to different photometric conditions. Additionally, we present an effective and robust multi-view object segmentation approach as a valuable byproduct of our pipeline. Extensive experiments demonstrate that our method significantly surpasses existing frameworks in maintaining content consistency across views and enhancing visual quality.

## CCS CONCEPTS

• **Computing methodologies → Reconstruction**.

## KEYWORDS

Visual editing, Image-based rendering, Radiance field, Multi-view consistency.

## 1 INTRODUCTION

Radiance Fields (RFs), such as Neural Radiance Fields (NeRF) [29] and 3D Gaussian Splatting (3D-GS) [17], are revolutionizing 3D scene representation and enhancing the realism of novel view synthesis. This technology holds great promise for Virtual and Augmented Reality (VR/AR), film production, and video game development. However, a significant challenge with the practical application of RFs is the difficulty of content modification, such as object removal. In implicit RF models (*e.g.*, NeRF), direct editing is challenging because scenes are encoded within neural network weights, which restricts precise user control over specific objects. In contrast,

*ACM MM, 2024, Melbourne, Australia*

© 2024 Copyright held by the owner/author(s). Publication rights licensed to ACM.
ACM ISBN 978-x-xxxx-xxxx-x/YY/MM
https://doi.org/10.1145/nnnnnnn.nnnnnnn

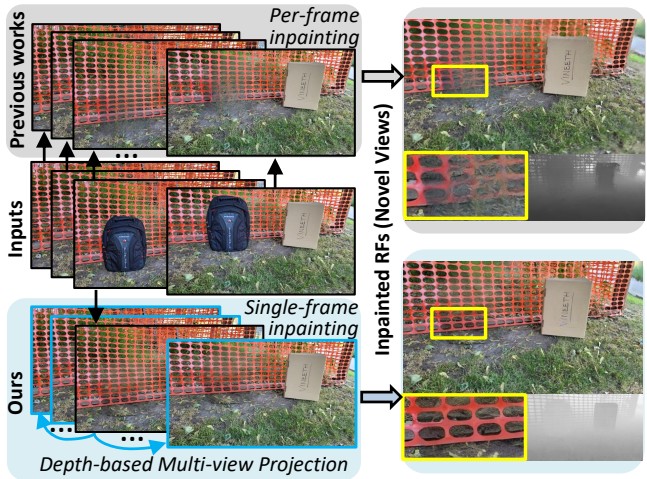

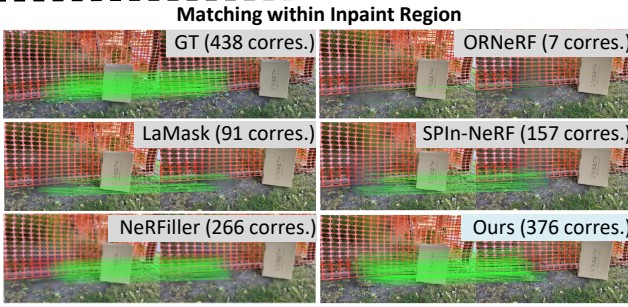

**Figure 1: An illustration of our radiance field (RF) inpainting pipeline. Unlike conventional methods that inpaint on a per-frame basis, our approach inpaints a single reference image and applies depth-based projection to seamlessly extend the modifications across multiple views. We show that our method not only enhances the quality of inpainted RF scenes but also significantly improves correspondence between different perspectives.**

explicit RF models (*e.g.*, 3D-GS) encounter difficulties with unclear surface definitions, which hinder accurate object segmentation and complicate the editing process. Therefore, achieving high-quality modifications in RFs is nontrivial.

3D scenes represented by RFs can be derived from sparse 2D images. To remove objects from these scenes, 2D inpainting methods are commonly used. Current works [30, 40, 59] typically begin with the creation of a multiview mask via image/video segmentation, which identifies the areas needing removal across different views. These specified areas are then independently inpainted for each view. However, these approaches have several shortcomings. The primary issue is the lack of consistency in object appearance and texture across different frames, as each frame is inpainted independently. This can lead to visual artifacts and unreliable scene

geometry. Furthermore, achieving consistent segmentation itself is challenging with sparse inputs. Image-based segmentation methods can exhibit large variability between frames, while video-based segmentation methods struggle with images captured from infrequent or diverse angles. The limitations of these existing approaches highlight a significant gap in our ability to edit RF scenes without compromising their inherent realism and coherence.

In this paper, we proposed a novel RF inpainting method designed to maintain view consistency in object removal within 3D scenes (Fig. 1). This method simplifies the editing process by inpainting just a single, centrally-located reference image rather than multiple individual views. We then utilized depth-based projections to map the inpainted results from the reference view to other training views, effectively reducing inconsistencies commonly seen in per-frame inpainting and maintaining content consistency in the masked regions. Another key advantage of this method is the ability to utilize more advanced 2D inpainting techniques, such as diffusion-based generative models [11, 36]. These models produce highly realistic and detailed textures but typically falter in multi-view inpainting due to their stochastic nature. By applying these advanced techniques exclusively to the reference view, we can harness their strengths for high-quality inpainting while maintaining consistency across multiple views.

To effectively handle realistic variations in lighting and viewpoint, our pipeline strategically adjusts the appearance of projected views. Traditional depth-based projection methods transfer RGB values directly from the reference to the target regions under the assumption of uniform lighting conditions. However, this assumption often fails in real-world applications due to varying lighting and perspective shifts. To overcome this, we generate multiple directional variants of the inpainted reference image, each tailored to a different target direction. This is achieved by querying the reference view with color representations adjusted for each target direction. During the projection phase, we select the corresponding variant according to the target view, thus preserving both structural and view-dependent consistencies.

Another valuable byproduct of our pipeline is the depth-based multi-view segmentation method, which efficiently and robustly provides consistent masks across views. In summary, our proposed approach maintains consistency in both masks and inpaintings across all views, ensuring compatibility with various RF models, such as NeRF and 3D-GS. We have demonstrated the effectiveness of our method using these models, highlighting its versatility and potential to enhance RF scene editing capabilities.

The contributions of this paper are summarized as follows:

(1) A novel RF inpainting method that requires inpainting only one reference view, significantly enhancing efficiency and consistency across multiple views.
(2) A directional variants generation module adjusts the appearance of projected views to enhance the photorealism of the synthesized views.
(3) The development of a fast and robust multi-view segmentation approach to facilitate precise location and removal of objects across views.

## 2 RELATED WORK

### 2.1 Image Inpainting

Image inpainting is a problem that has been long studied in the field of computer vision [34]. Initial approaches to image inpainting primarily relied on the low-level features of damaged images, involving methods based on Partial Differential Equations (PDE) [1, 2, 44] and patch-based techniques [7, 10, 12]. Nowadays, deep learning based image inpainting methods has taken a dominate position. As mentioned by [34], deep learning based inpainting method can be classified as 1) deterministic image inpainting and 2) stochastic image inpainting. Given a image and its corresponding mask, deterministic image inpainting methods only produce an inpainting result, whereas stochastic image inpainting approaches are capable of generating several plausible outcomes through a process of random sampling.

As for deterministic methods, researchers often utilize three types of framework: single-shot, two-stage, and progressive methods. Single-shot methods [4, 25, 49, 53, 63] utilize an end-to-end generator network to output the inpainting result. Two-stage methods [37, 42, 55, 61, 62] consists of two generators and follows a coarse-to-fine strategy. The progressive methods [8, 22, 23, 66, 67] utilize multiple generators to inpaint the masked region in the given image in a iterative manner.

For stochastic methods, we can divided them into VAE-based methods [9, 18, 32, 45, 69, 70], GAN based methods [6, 16, 26, 68, 71], flow-based methods [5, 35, 48], MLM-based methods [46, 64] and Diffusion model-based methods. As diffusion model [11] has gained increasing popularity in recent years, latent diffusion models (LDMs) [24, 28, 54] has become the dominant method in the field of image inpainting.

In our work, we select diffusion model-based methods as they can produce more reasonable and photo-realistic inpainting results. Due to the nature of our work, we don't need to care about the stochastic property of diffusion models. While most of the previous works utilize LaMa [43], which is a deterministic method as they didn't explicitly handle the inconsistent inpainting issue.

### 2.2 3D Editing

With the emergence of NeRF and Gaussian Splatting, many excellent works [14, 15, 20, 21, 27, 31, 47, 57, 60, 65], have sprung up in the field of 3D scene editing. Some works [65] focus on the editing of the explicit geometry after training a NeRF. Peng et al. and Xu et al. [33, 56] try to make NeRF deformable and capable of animating general objects. Many works also put emphasis on object-centric editing. Wu et al. [52] proposed ObjectSDF, which is an object-compositional neural implicit representation, it is able to represent the surface of each object and the entire scene accurately. Yang et al. [57] proposed an object-compositional neural radiance field that is able to apply simple transformation and manipulation to the objects in the scene.

For object removal, there are also several works appear in recent years SPIn-NeRF [30], NeRF-In [40], Removing Objects From Neural Radiance Fields [51] and NeRFiller [50] demonstrate the ability to remove objects in NeRF. OR-NeRF [59] proposed a faster multi-view segmentation method and leverage TensoRF [3] to boost the rendering quality. Point'n Move [13] is able to handle object

removal in 3D Gaussian Splatting. All the object removal methods mentioned above are based on image editing, and then utilize inpainted images to train a inpainted radiance field. So the inconsistency of inpainting results in different views is a crucial problem to be solved. However, none of them have handled this issue perfectly. NeRF-In utilize pixel-wise MSE loss to simply supervise the content in the masked region, and does not have any further approach to deal with inconsistent inpainting result. SPIn-NeRF and OR-NeRF loosen the constrain provided by pixel-wise MSE loss and utilize perceptual loss to guide the optimization in the masked region to produce a more visually smooth result, but as viewpoint changes, the content in the inpainted region may still change slightly.

One recent work, NeRFiller [50], proposed to use Grid Prior (tile multiple images into a grid) for generating consistent inpainting images and propagate the inpainted part into the entire 3D scene in a iterative dataset update manner. According to our experiment, they did improve some 3D consistency, but the rendering quality is not satisfying. The reason for this is that they still need multiple times of inpainting. Though the consistency in maintained within each inpainting, there still exists inconsistency between different inpainting attempts.

In contrast, our method can maintain cross-view consistency during the inpainting process by explicitly projecting the generated content into all the training images. And due to the pre-mentioned drawback, the above approaches (except NeRFiller) cannot leverage advanced image inpainting methods e.g. Stable Diffusion to generate more photo-realistic results in complex environments. Because of the stochastic nature of diffusion model, even the input images are in the same environment, you can hardly get similar inpainting results. Since in our method we only need to inpaint one reference image, we do not have to deal with this issue.

## 3  PRELIMINARY: RADIANCE FIELDS

We demonstrate the effectiveness of our method using both implicit RF (*i.e.*, Neural Radiance Fields (NeRF) [29]) and explicit RF (*i.e.*, 3D Gaussian Splatting (3D-GS) [17]).

**NeRF.** NeRFs represent 3D scenes as Radiance Fields that maps the 3D coordinate $x$, $y$, $z$ and the viewing direction $\theta$, $\phi$ to color $c$ and density $\sigma$. To get the color of a pixel, a ray will be shot through the pixel and then multiple points on the ray will be sampled. The color and density of each sampled point will be predicted by an MLP. Finally, volume rendering will be used to accumulate these sampled colors and render the pixel color $\widehat{C}$:

$$\widehat{C} = \sum_{i=1}^{N} T_i \left(1 - \exp\left(-\sigma_i \delta_i\right)\right) c_i,$$

where $T_i = \exp\left(-\sum_{j=1}^{i-1} \sigma_j \delta_j\right)$ is accumulated transmittance to the current sample point $t_i$, representing the probability that light travels from the camera to the point without hitting any other particles, and $\delta_i$ is the distance between adjacent sample points on the ray. $c_i$, $\sigma_i$ correspond to the color and density at $t_i$. Reconstruction loss between ground truth color $C$ and the predicted color $\widehat{C}$ is calculated to supervise the training process of NeRF.

**3D-GS.** 3D Gaussian Splatting utilizes a set of 3D ellipsoids to explicitly represent a scene. Each ellipsoid is modeled by an anisotropic

3D gaussian, which is parameterized by a center point $x$ (mean of gaussian) and a covariance matrix $\sum$. The color of each gaussian is parametrized by spherical harmonics.

During the rendering process, 3D gaussians are first projected into image plane as 2D gaussians. Then the color of each pixel is calculated through the alpha-blending process over the points overlapping that pixel.

$$\widehat{C} = \sum_{i \in \mathcal{N}} c_i \alpha_i \prod_{j=1}^{i-1} \left(1 - \alpha_j\right),$$

where $c_i$ is the color of each point calculated through spherical harmonics, and $\alpha_i$ is the opacity calculated from the covariance matrix $\sum'$. The rendered color is used to calculate the reconstruction loss with the ground truth color to optimize the 3D gaussians.

## 4  METHOD

In this part, we will describe our proposed method to maintain cross-view consistency for object removal in RFs and dive deeper into the details of each step in the following sections.

Our entire pipeline is shown in Fig. 2. We first select a camera with the least average distance on SO(3) manifold to all other cameras in the training data as the reference view. Then, the reference view is processed to get the mask $M_r$, inpainted reference view $I_r$, the depth map $D_r$ of $I_r$ (section 4.1). We then utilize depth-based projections to transfer the inpainted results from the reference view to other views, generating multi-view segmentation and inpainting results (section 4.2). Finally, an inpainted Radiance Field will be trained using the set of inpainted training images with the following reconstruction loss:

$$\mathcal{L}_{\text{rec}} = \sum_{k=0}^{N} \left\| \widehat{I_k} - I_k \right\|^2,$$

where $I_k$ is the inpainted images via multi-view projection, $\widehat{I_k}$ is the generated results of inpainted RF, and $N$ is the number of images.

### 4.1  High-quality Single-view Processing

We initiate our methodology by selecting a reference camera pose from the training dataset; this camera pose is identified as having the minimal average distance to all other poses on the SO(3) manifold. The processing of the chosen reference view involves three key steps: masking, inpainting, and depth estimation, yielding three outputs: the mask $M_r$, the inpainted image $I_r$, and the depth map $D_r$, respectively. These outputs are crucial for subsequent multi-view projection and inpainting tasks.

**Mask Generation and Image Inpainting.** To generate the mask $M_r$ of the reference image, we employ the Segment Anything Model (SAM) [19], an advanced off-the-shelf model known for its efficiency and accuracy in image segmentation. For the inpainted image $I_r$ of the reference view, we leverage a pretrained 2D inpainting model (*i.e.*, Stable Diffusion [36]) to fill the masked region with realistic texture and fine details.

**Depth Map Estimation and Alignment.** Generating the depth map $D_r$ presents unique challenges, particularly regarding accuracy and smoothness. Previous approaches [30, 59] have relied on the trained RFs (*e.g.*, NeRF and 3D-GS) to derive depth information.

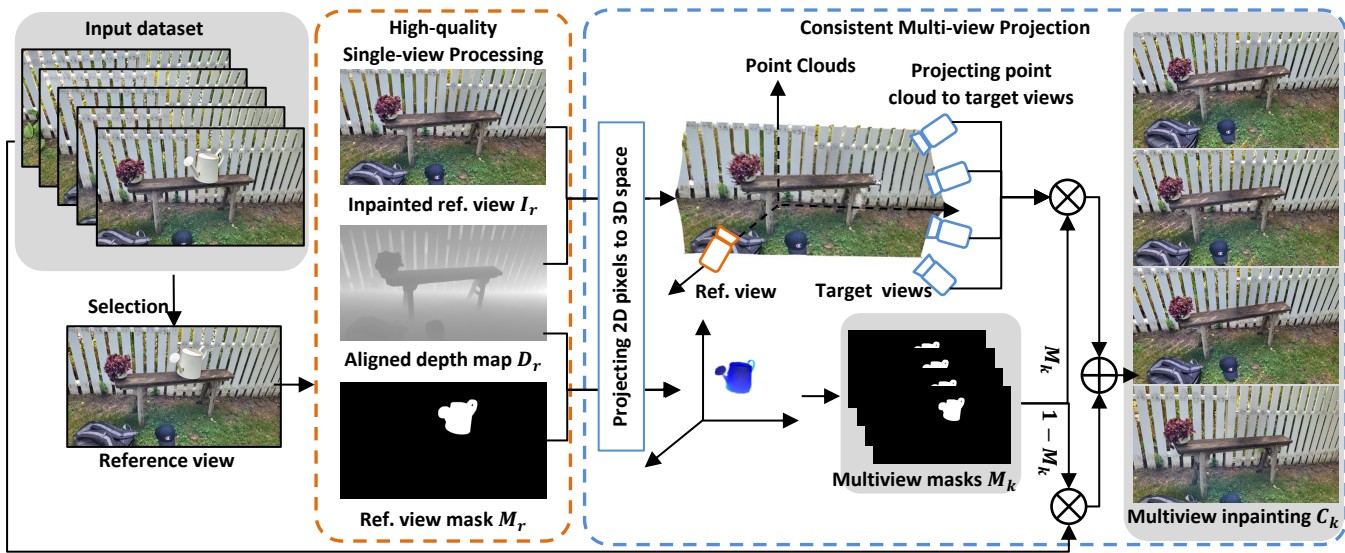

**Figure 2: An overview of our method: we initiate our methodology by selecting a reference camera pose from the training dataset; this camera pose is identified as having the minimal average distance to all other poses on the SO(3) manifold. The processing of the chosen reference view involves three key steps: masking, inpainting, and depth estimation, yielding three outputs: the mask $M_r$, the inpainted image $I_r$, and the depth map $D_r$, respectively. These outputs are then used for multi-view projection, yielding a set of inpainted images from multiple views. Finally, an inpainted Radiance Field will be trained using these inpainted images.**

However, RFs are sensitive to noise in the input data, which can degrade the depth map with artifacts and uneven surfaces. Such degradation will lead to irregular projection gaps during the multi-view projection process that rely on depth information.

To achieve precise and coherent depth information for projection, we start by estimating the depth with a monocular depth estimation method (*i.e.*, Depth-Anything [58]), producing an initial smooth depth map $D_{init}$. To resolve the scale ambiguity inherent in monocular depth estimation, we align $D_{init}$ with sparse depth data $D_{col}$ generated from the Structure-from-Motion (SfM) library COLMAP [39] to accurately scale the depth.

We approach the depth alignment between $D_{col}$ and $D_{init}$ as a least square problem, aiming to minimize the cost function:

$$\mathcal{L}_{\text{align}} = \sum_{i \in D_{col} \odot (1-M_r)} D_{col}^i - (a \cdot D_{init}^i + b),$$

where $D_{col}^i$ and $D_{init}^i$ represent the depth of the $i^{th}$ pixel in $D_{col}$ and $D_{init}$, respectively, while $a$ and $b$ are the scale coefficients. To ensure accuracy around the object, we omit depth pixels both within the mask region $M_r$ and those significantly distant from it. Once the optimal scale coefficients (*i.e.*, $a^*$ and $b^*$) are determined, the final aligned depth $D_r$ is calculated as:

$$D_r = a^* \cdot D_{init} + b^*.$$

This depth estimation and alignment strategy ensures our depth map $D_r$ is not only accurate but also exhibits a smooth gradient, which is essential for error-free multi-view projection and inpainting workflows.

## 4.2 Multi-view Consistent Inpainting

**Inpainting via Projection.** Inspired by depth image-based rendering (DIBR) techniques, we utilize depth-based projections to transfer the inpainted results from the reference view to other views, after processing the single reference view. This approach addresses the common inconsistencies found in per-frame inpainting and ensures content consistency within the masked regions.

Upon obtaining the inpainted reference image $I_r$ and its corresponding depth map $D_r$, our goal is to project $I_r$ onto a target view $k$, generating the inpainted image $I_k$. We start by backprojecting each 2D pixel in $I_r$ into 3D space to create a point cloud $\mathbf{c}_r$ using its depth information. Specifically, the $i^{th}$ pixel in $I_r$, denoted as $I_r^i$, corresponds to a point $\mathbf{c}_r^i$ in 3D space. The coordinates of each point $\mathbf{c}_r^i$ are calculated as follows:

$$\mathbf{c}_r^i = \begin{bmatrix} X_r^i \\ Y_r^i \\ Z_r^i \end{bmatrix} = D_r(u_r^i, v_r^i) \cdot K^{-1} \cdot \begin{bmatrix} u_r^i \\ v_r^i \\ 1 \end{bmatrix},$$

where $K$ is the camera intrinsic matrix and $(u_r^i, v_r^i)$ are the coordinates of the $i^{th}$ pixel in inpainted reference image $I_r$. $D_r(x, y)$ represents the depth value at position $(x, y)$ in the depth map $D_r$.

Following this, we project the point cloud $\mathbf{c}_r$ to the new viewpoint $k$ through a relative transformation matrix $T$ between the reference and the target viewpoints:

$$c_k^i = \mathbf{T} \cdot c_r^i,$$

where $c_k^i$ is the projected point in view $k$. The coordinates of points in the target image space are then calculated as:

$$\begin{bmatrix} u_k^i \\ v_k^i \\ 1 \end{bmatrix} = K \cdot \frac{1}{Z_k^i} \cdot c_k^i,$$

where $Z_k^i$ is the depth of point $c_k^i$.

The pixel values in the target view's masked region are then replaced by the corresponding projected pixel values:

$$I_k(u_k^i, v_k^i) = I_r(u_r^i, v_r^i).$$

After projection, another crucial task is to handle the projection gaps due to occlusions. We utilize LaMa [43] to inpaint these small and regular gaps, resulting in a set of refined projection results $\{I_k\}$, $k = 0, 1, 2, \cdots, N-1$, where $I_k$ represents the projected inpainting result from $I_r$ to view $k$.

Similarly, given the mask $M_r$ of the reference view along with the depth information $D_r$, we can project $M_r$ onto target view $k$ and get the corresponding mask $M_k$. This method enables the automatic generation of robust and view-consistent segmentations across multiple views.

**View Dependent Effect.** Directly projecting and propagating the inpainting result from the reference view to all the other training views has an assumption of photometric consistency. However, this assumption often fails in real-world scenarios, where lighting and viewpoint variations are common. To address these challenges and better adapt to realistic variation, we strategically adjust the appearance of projected views to better match their target settings. The approach involves generating multiple directional variants of the inpainted reference image, each tailored to a specific target direction. During the projection phase, we select and utilize the variant that best corresponds to the target view.

To generate these directional variants, we first train a Radiance Field with the original training set before inpainting. Then, we extract the view-dependent appearance encoded in the trained RF representation. This is done by maintaining the camera's viewpoint as fixed at the reference view while varying the queried viewing directions with the target views. This process results in $N-1$ directional variants of the reference image, each reflecting different lighting conditions. We then utilize Stable Diffusion to inpaint these reference views. Empirical evidence suggests that the content generated by Stable Diffusion maintains geometric consistency under minor variations in lighting conditions. By leveraging this property, we are able to accurately generate and project these variants across different views, ensuring that the adjustments align well with the varying conditions of each target view.

**Depth-Based Occlusion Correction.** During the projection process, multiple points from the reference point cloud $\mathbf{c}_r$ may be projected onto the same pixel in the target view. Thus, we need to maintain a Z-buffer to ensure that the points with small depth values will remain on the image plane.

Besides, some pixels primarily occluded may be revealed at the surface accidentally. The reason why this happens is that the inpainted reference view may have some content that should be occluded in the target view. They are now exposed at the surface

---

**Algorithm 1** This pseudo-code describes how to utilize Z-buffer and Depth Prior to help deal with the occlusion and de-occlusion issue during depth-based projection process.

---

**Require:** $\mathbf{c}_r, \mathbf{D}_r, \mathbf{T}, \mathbf{D}_{prior}$
  $z\_buf[1 \ldots n] \leftarrow$ new Array$(n)$
  **for** $i = 1$ to $n$ **do**
    $z\_buf[i] \leftarrow \infty$
  **end for**
  **for** $k = 1$ to $n$ **do**
    $u_k^i, v_k^i, Z_k^i \leftarrow \text{DIBR}(\mathbf{c}_r, \mathbf{D}_r, \mathbf{T})$
    **if** $0 \le u_k^i < w$ **and** $0 \le u_k^i < h$ **and** $Z_k^i < z\_buf[i]$ **then**
      **if** $\mathbf{D}_{prior}(u_k^i, v_k^i) - Z_k^i > \epsilon$ **then**
        $I_k(u_k^i, v_k^i) \leftarrow I_r(u_r^i, v_r^i)$
        $z\_buf[i] \leftarrow Z_k^i$
      **end if**
    **end if**
  **end for**

---

because the foreground content that should cover them is not available in the reference view, which means they are in the projection gap and thus not available.

To deal with this issue we introduce depth prior during the projection process. Briefly speaking, we first estimate the depth of each target view as described in section 4.1. Then during the projection process, we utilize the estimated depth map as a depth prior, and reject any pixel that has a depth larger than the corresponding depth prior. The detailed algorithm to deal with occlusion and de-occlusion issue is shown in algorithm 1 as pseudo-code.

## 5 EXPERIMENTS

### 5.1 Dataset and Implementation Details

All the following experiments are accomplished based on the SPIn-NeRF dataset [30], which was designed specifically for 3D inpainting. SPIn-NeRF dataset contains 10 scenes, including both indoor and outdoor scenarios. Within each scene, there are 60 training images including the unwanted object, and 40 ground truth images with the unwanted object removed. The dataset also provide human annotated segmentation masks for each training images.

We run both vanilla NeRF and 3D-GS based on our inpainting results, without additional modification on loss function or training procedure to show the effectiveness of our proposed method.

For the initialization of 3D-GS, we first remove the unnecessary points in the masked area from the sparse point cloud generated by colmap, and then leverage the aligned depth estimation results produced in section 4.1 to serve as the initialization for the mean of 3D Gaussians inside the masked region.

### 5.2 Radiance Field Inpainting

For the quantitative comparison on Radiance Field Inpainting, we report the average peak signal-to-noise ratio (PSNR), the average learned perceptual image patch similarity (LPIPS), and the average Fréchet inception distance (FID) between the rendered test view and the ground truth test image provided by the SPIn-NeRF dataset.

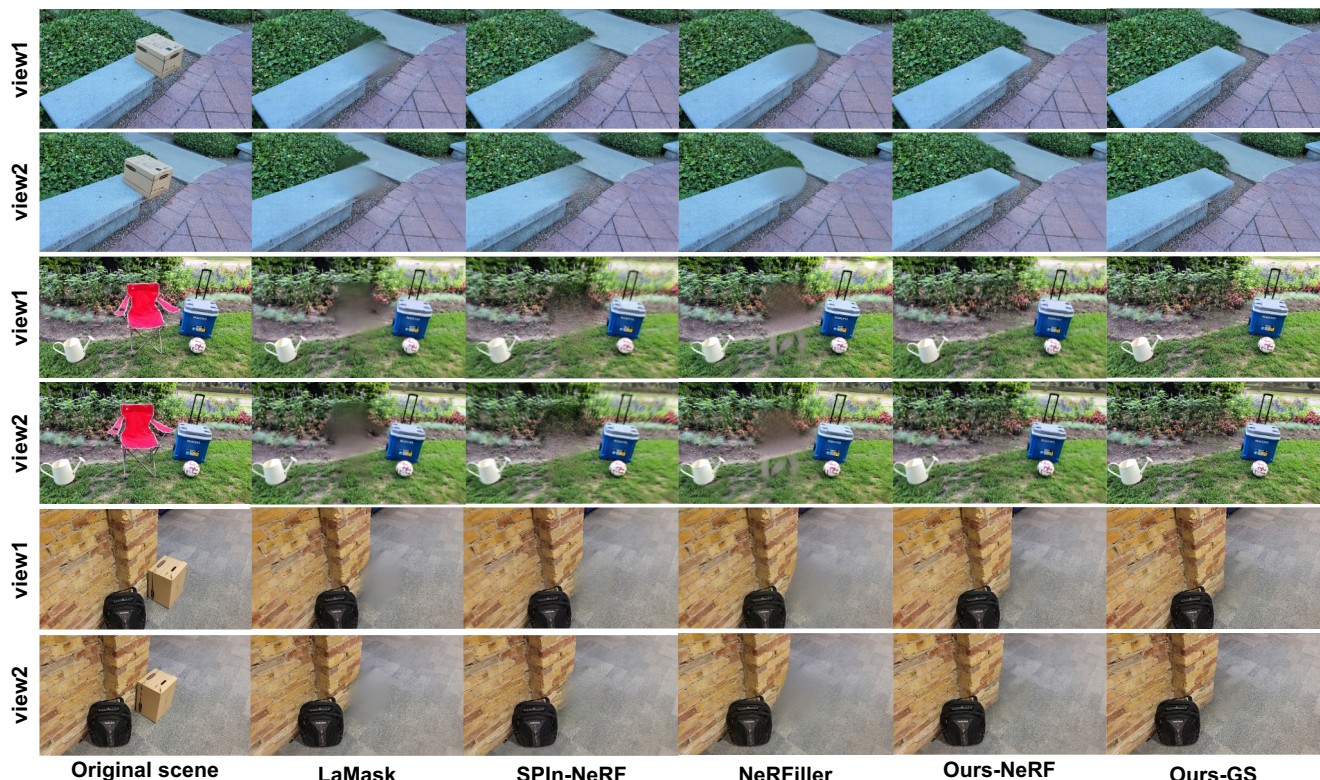

**Figure 3: Qualitative comparison between our methods and baseline methods. For each scene, we show images from two different views to compare both rendering quality and cross-view consistency.**

Note that the ground truth images are only used for evaluation, and are not required during training. Our baselines are the following:

(1) **LaMask** - Inpainting all the training images with LaMa [43], and train a vanilla NeRF without any other techniques based on these inpainted images.

(2) **SPIn-NeRF** [30] - Based on LaMask, utilize depth inpainting as depth supervision and apply perceptual loss, LPIPS within the mask region to solve the blurry issue caused by inconsistent inpainting.

(3) **OR-NeRF (TensoRF)** [59] - Enhanced version of SPIn-NeRF, using TensoRF instead of vanilla NeRF.

(4) **NeRFiller** [50] - NeRFiller utilize grid prior (tile the input images into a grid and treat the entire grid as a single inpainting target) to generate more consistent inpaintings. And proposed an iterative 3D scene optimization method to maintain global 3D consistency.

The quantatitive results are shown in Table 1. Our inpainting method trained with Gaussian Splatting (Ours-GS) achieves the best performance in terms of LPIPS and FID score, and Ours-NeRF outperforms all the other models in PSNR. It is worth mentioning that though Ours-NeRF utilizes vanilla NeRF as backend, it still achieve competitive or even better results compared with ORNeRF (TensoRF backend) and NeRFiller (Nerfacto backend). We also show some qualitative comparison in Fig. 3.

| Methods | PSNR ↑ | LPIPS ↓ | FID ↓ |
|---|---|---|---|
| SPIn-NeRF | 20.63 | 0.39 | 68.23 |
| LaMask | 20.27 | 0.41 | 63.06 |
| ORNeRF-TensoRF | 18.53 | 0.25 | 48.28 |
| NeRFiller | 19.71 | 0.37 | 72.79 |
| Ours-GS | 20.22 | **0.21** | **35.69** |
| Ours-NeRF | **20.82** | 0.38 | 47.79 |

**Table 1: Quantitative comparison of our inpainting method with ground truth object masks**

## 5.3 Multi-view Consistency

**Inpainting Consistency.** In this section, we evaluate the multi-view consistency of our methods against the baseline approaches. We apply widely used off-the-shelf image feature matching methods LoFTR [41] and SuperGlue [38] to check the number of correspondence between the image pairs rendered by ours and baseline methods. The comparison results are shown in Table 2. For both feature matching methods, we randomly sample 100 images pairs to calculate the correspondence and only the matchings within the masked region are taken into consideration. For LoFTR, we only calculate the correspondence with confidence level higher that 0.95. We use pretrained weight "indoor" for scene 9, book and trash

| Methods | LoFTR | SuperGlue |
|---------|-------|-----------|
| SPIn-NeRF | 154.03 | 19.44 |
| LaMask | 105.79 | 23.11 |
| ORNeRF-TensoRF | 34.48 | 18.07 |
| NeRFiller | 201.34 | 22.94 |
| Ours-GS | 283.52 | 40.25 |
| Ours-NeRF | **319.04** | **64.40** |

Table 2: Number of correspondence found between pairs of rendered images. A higher correspondence value indicates better geometry consistency.

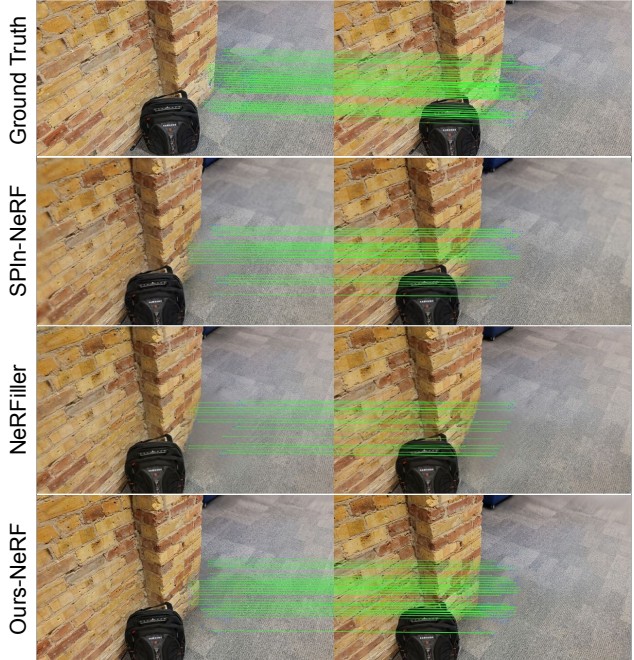

Figure 4: Visualization of feature matching results within the masked region. Ground Truth, SPIn-NeRF, NeRFiller, and Ours-NeRF have number of matchings 329, 193 , 84 and 324 respectively. The original scene picture is shown in Fig. 3

and all the other scenes are evaluated with "outdoor" weight. As for quantitative results, our inpainting approached outperform the baseline methods in both of the matching method.

We also visualize the matching results of LoFTR in Fig. 4 for comparison. The first row in Fig. 4 shows the matching result between two ground truth images with unwanted objects removed provided by the SPIn-NeRF dataset.

**Mask Consistency.** The mask consistency across different views is also quite crucial in the Radiance Field editing process. Inconsistent masks will cause inconsistent inpainting and thus break the 3D consistency. Here, we compare our method with two segmentation methods proposed by OR-NeRF to demonstrate our mask consistency. OR-NeRF proposed two segmentation methods 1) point prompt based and 2) text prompt based. The point prompt based

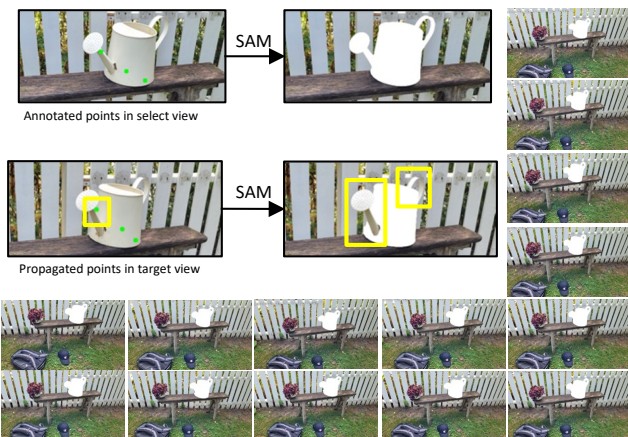

Figure 5: Failure case of OR-NeRF (point prompt) is on the upper left corner. The first row shows the manually annotated point prompts in a selected view and its corresponding mask generated by SAM. The second row shows the propagated point prompts to another view and its corresponding mask. We can see that one of the propagated point prompt does not lay on the expected region and thus the generated mask is not completed.

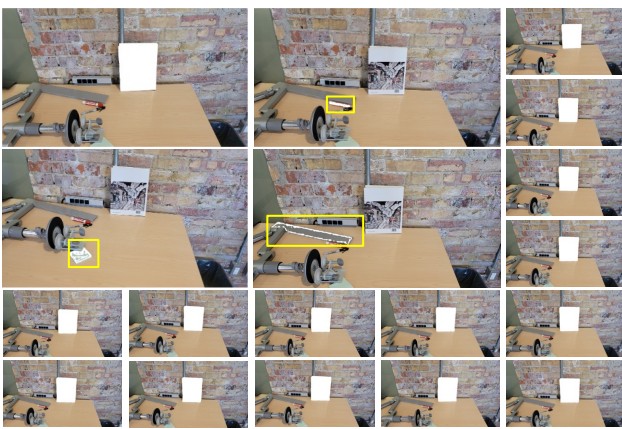

Figure 6: Failure case of OR-NeRF (text prompt) is on the upper left corner. We use the text prompt "book" to do the segmentation. We can see that SAM may incorrectly segment the pen, sticky note and the metal bar on the table as "book".

one requires manually annotating some points on a selected 2D image and utilize the sparse point cloud generated by colmap to spread the point prompt to all the other views. The text prompt one uses a single text prompt for SAM to just the segmentation result for all the images. However, both of them have some drawbacks. For point prompt, not all the annotated points can be found in the point cloud, and thus they need to find a closest point as replacement, which may cause an offest during propagation. For text prompt, it is quite hard to find a universal prompt that works for all the images.

| Method | Acc. ↑ | IoU ↑ | Dice ↑ |
|---|---|---|---|
| SPIn-NeRF | 98.91 | 91.66 | - |
| OR-NeRF (text) | 97.78 | 72.75 | 84.26 |
| OR-NeRF (points) | **99.63** | 94.07 | 96.84 |
| Ours | 99.48 | **94.27** | **96.98** |

**Table 3: quantitative comparison between our proposed multi-view segmentation methods and the baseline methods**

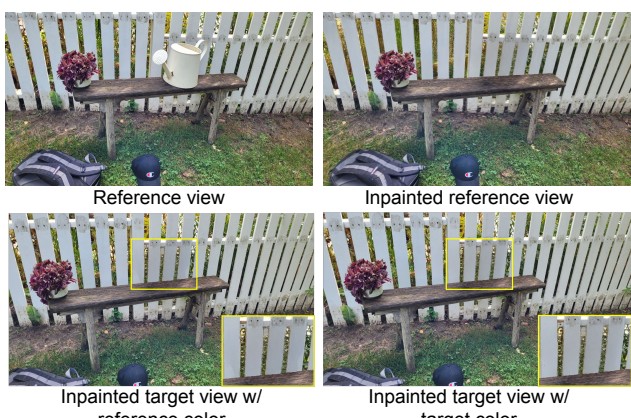

**Figure 7: Ablation study for view-dependent effect. reference view of different lighting conditions. The first row shows the selected reference view and its corresponding inpainting result. The second row shows the inpainting result w/ and wo/ using the lighting variant reference view.**

We show some failure cases of OR-NeRF and also our segmentation results over the same scene in Fig. 5 and Fig. 6 to proof the mask consistency of our proposed method. Our results are shown at the periphery of these two figures.

We then quantitatively compared our depth projection based multi-view segmentation method with the MVSeg model provided by SPIn-NeRF and the points/text prompt based multi-view segmentation method propsed by OR-NeRF. We report average accuracy, intersection over union (IoU) and Dice score between the human-annotated ground truth mask and the mask predicted by different approaches, the numerical results are shown in Table 3. For SPIn-NeRF, as they didn't report Dice score in their paper and the code for MVSeg is currently not available, we just leave it blank.

## 5.4 Ablation Study

**View-dependent Effect.** We rendered multiple directional variants of reference views as indicated in section 4.2. In this ablation study, we show the effectiveness of this module. Fig. 7 shows a comparison between the inpainting result of the target view projected by the original reference view and the result projected by the reference view with lighting variation. We can see that if we directly project the inpainted area to the target view without changing the

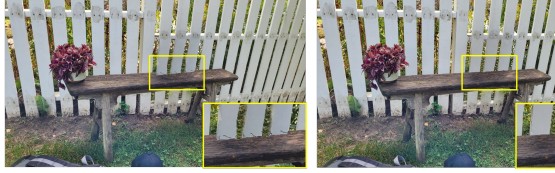

**Figure 8: Ablation study for depth-based occlusion correction.**

| Methods | PSNR ↑ | LPIPS ↓ | FID ↓ |
|---|---|---|---|
| Ours-GS (GT mask) | 20.22 | 0.21 | 35.69 |
| Ours-GS (Our mask) | 20.14 | 0.21 | 35.32 |
| Ours-NeRF (GT mask) | 20.82 | 0.38 | 47.79 |
| Ours-NeRF (Our mask) | 20.68 | 0.39 | 48.15 |

**Table 4: Quantitative comparison between the Radiance Field inpainting results using human-annotated mask and the mask generated by our proposed segmentation methods.**

lighting condition, it will result in an obvious contour around the inpainting region. After applying the target color to the reference view, this phenomenon is eliminated.

**Depth-Based Occlusion Correction.** As claimed in section 4.2, z-buffer and depth prior are used to solve the issue of occlusion and de-occlusion of projected points. Here we visualize the above mentioned issue and show the improved inpainting result with depth-based occlusion correction. From the left image in Fig. 8, we can see that some part of the barrier ought to be occluded by the bench is now revealed at the surface. After applying the depth-based occlusion correction, the occlusion relationship between the bench and the barrier is corrected.

**Multi-view Segmentation.** We also quantitatively compare the RF Inpainting results using masks generated with our segmentation method and the ground truth masks provided by the dataset (Table 4). It shows that using the masks generated by our method only results in subtle performance degradation in RF inpainting.

## 6 CONCLUSION

Our work introduces a novel RF editing pipeline designed to overcome the 3D inconsistency issue during 3D object removal. By employing a strategy of inpainting a single reference image followed by depth-based projection, our method efficiently extends the inpainted effects across multiple views, thereby minimizing the inconsistencies observed with per-frame inpainting approaches. Furthermore, we also accommodate view-dependent effect by adjusting observed colors based on the viewing direction, which is determined during the color querying phase. Through rigorous testing, we demonstrate that our method maintains content rationality and significantly improves the visual quality of RF scenes, which marks a substantial advancement over existing frameworks.

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
