# OpenReview forum: "View-consistent Object Removal in Radiance Fields"
_acmmm.org/ACMMM/2024/Conference — MM2024 Poster_

### Official Review · Reviewer_zNtY · 2024-05-06

**Rating:** 5
**Confidence:** 4

**Summary:**

The paper is on the topic of editing radiance fields at the example of view-consistent object removal. The authors show that&mdash;independent of whether explicit (like Gaussian Splatting) or implicit (like NeRFs) radiance field representations&mdash;are used the can remove objects while maintaining the consistency of the radiance fields.
To achieve the goal they inpaint a single (central) image and diffuse the created content to other views, exploiting characteristics like direction dependent radiation resp. coloring. The results are very convincing.

**Strengths:**

The paper shows a complete pipeline for radiance field based object removal, aka they produce an output radiance field from which objects&mdash;which have been masked manually&mdash;are removed.
The strength of the paper is in plausibly exploiting features of the radiance field to propagate single-view inpainting results to other viewing directions. The projection is done DIBR-based, but guided by different directional variants taken from the original radiance field (so that the projected content outside the mask indeed faces a directionally correct warping, enforcing the inpainting to also be directionally correct). In addition the depth maps are used to correct occlusions (multiple points of the reference view might end up in the same pixel of the projected view, but only the occluding one shall be maintained, not the occluded ones).
Even though the improvement of the state-of-the-art with respect to classical metrics like PSNR is small (in the range of 0.2dB), the gain in perceptual quality (LPIPS) and distribution similarity (FID) is significant (table 1). It, however, becomes visible that explicit radiance field representations like Gaussian Splatting seem to work significantly better than NeRFs.
As a by-product the scheme provides a multiview segmentation, aka view dependent masks.

**Limitations:**

The paper uses the SPIn-NeRF dataset which contains only a limited number of scenes (10). On the other hand, the algorithm itself requires the same input as NeRFs (so essentially works on multiview images plus camera-poses, not on radiance fields).
Figure 3 shows images which seem to have been shot by the authors (I don't know the exact scenes within SPIn-NeRF); the authors should clearly summarize the images taken for training validation and the ones taken for out-of-data testing (show a mosaic of the central views).
Figure 3 should also include some zoom-ins into the interesting regions around the removed object (or even only depict the bounding box of the removed object).

**Suitability:**

3

---

### Official Review · Reviewer_6GNs · 2024-05-21

**Rating:** 4
**Confidence:** 3

**Summary:**

This work proposes using a single reference view and multiple view lighting constraints for view-consistent object removal.

**Strengths:**

The comparison results demonstrate the proposed method's excellent performance, and the visual outcomes are impressive.

The writing is clear and well-articulated.

**Limitations:**

1. The reference view is selected based on the minimal average distance to all other poses on the SO(3) manifold (Line 310-312). Are there alternative methods for choosing the reference view, and do different options significantly impact the final results? What would the outcomes look like if a reference view were chosen randomly?

2. This work utilizes Stable Diffusion to inpaint the reference views (Line 505-507), which may introduce some inconsistency between different views. If the inconsistency is too great, does this influence the effectiveness of the method?

3. Would it make more sense to add new objects to an area instead of object removal? I'm not sure what's the point of removing only objects in Radiance Fields? What are the application scenarios?

4. I suggest including more citations and discussions of relevant MM conference papers on 3D to enhance the article's relevance for this conference. For example, "Geometry-Aware Reference Synthesis for Multi-View Image Super-Resolution" and "Space-Angle Super-Resolution for Multi-View Images."

**Suitability:**

2

---

### Official Review · Reviewer_ASis · 2024-05-27

**Rating:** 3
**Confidence:** 3

**Summary:**

This paper proposes a novel RF inpainting method, introducing an RF editing pipeline that requires inpainting only a single reference image. This inpainted reference image is subsequently projected across multiple views using a depth-based approach, thereby reducing inconsistencies.

**Strengths:**

1. The approach of inpainting a single reference image and projecting it across multiple views using depth information is innovative.

**Limitations:**

1. Could the authors provide details regarding the models' sizes, FLOPs, and other relevant metrics? The paper claims that inpainting a single reference view enhances efficiency. However, the use of Stable Diffusion for inpainting reference views is computationally intensive.
2. The proposed method inpaints only one reference view, with the results for other views derived from this inpainted view. How is the quality of the single-view inpainting ensured? Would multi-frame inpainting yield better results?
3. The experimental results presented are not sufficiently robust to demonstrate effectiveness. For instance, in Table 1, 'Ours-NeRF' shows only a 0.19dB improvement in the PSNR metric compared to SPIn-NeRF and performs worse in LPIPS and FID. Could these results be influenced by network fluctuations and random factors?

**Suitability:**

3

---

### Official Review · Reviewer_56Am · 2024-05-29

**Rating:** 1
**Confidence:** 4

**Summary:**

This paper introduces a Radiance Field (RF) editing pipeline that significantly enhances consistency by requiring the inpainting of only a single reference image.

**Strengths:**

The structure of this paper is complete.

**Limitations:**

1. The contribution is poor, as it only integrates the segmentation, inpainting, and reprojecting techniques. I don’t think that it is a simple, efficient, or novel method to remove objects in radiance fields. And I don’t deem that it is an editing strategy for radiance fields, which is a preprocessing step for radiance fields.

2. The experimental results are not convincing. The improvement is based on the segmentation, inpainting, the dense input views, and the selection of reference views. It doesn’t benefit from multi-view projection. Please report the results and conduct the ablation study of “segmentation + inpainting for per-frame before training RF”. It will be much more effective and efficient because without COLMAP and the depth supervision.

3. In lines 390 – 392, please interpret the necessity of this supervision in detail. Moreover, it would be better to report the result without this supervision.

4. In line 472, why the Zk is the depth of point ck rather than be calculated from reprojection?

5. In lines 477 – 482, how to ensure the reprojection only has the small and regular gaps?

6. If more than one pixels are reprojected to the same pixel, how to calculate the color?

7. Please report some conspicuous cases, which include the rendering results from multi-views after removing the object.

8. Please number each equation according to the ACMMM 2024 requirements.

**Suitability:**

2

---

### Meta-Review · Area_Chair_KwyP · 2024-07-04

**Recommendation:** Accept (Poster)
**Confidence:** 4

**Metareview:**

The paper presents an RF editing method that improves consistency by inpainting just one reference image, which is then projected across multiple views using a depth-based approach. This reduces inconsistencies seen with per-frame inpainting. The paper is clear and well-written. The results appear convincing, however, a few aspects need more attention and deliberation after the rebuttal. One reviewer's concern about the quality of the inpainted/ interpolated view is valid and is not properly commented on. I think, this point is important and requires more investigation. Secondly, the quality of the derived views is not discussed. Thirdly, a comprehensive time complexity analysis of the proposed method should be presented and if possible constructed with the compared methods.